# Integration of Micro-Nano-Engineered Hydroxyapatite/Biochars with Optimized Sorption for Heavy Metals and Pharmaceuticals

**DOI:** 10.3390/nano12121988

**Published:** 2022-06-09

**Authors:** Xin Zhao, Peiling Yuan, Ziyan Yang, Wei Peng, Xiang Meng, Jiang Cheng

**Affiliations:** 1Graduate Department, Civil Aviation Flight University of China, Guanghan 618307, China; zhaox@cafuc.edu.cn; 2Zhengzhou Key Laboratory of Low-Dimensional Quantum Materials and Devices, College of Science, Zhongyuan University of Technology, Zhengzhou 450007, China; 3School of Environmental and Municipal Engineering, North China University of Water Resources and Electric Power, Zhengzhou 450045, China; yangziyan@ncwu.edu.cn; 4Henan Province Key Laboratory of Water Pollution Control and Rehabilitation Technology, Henan University of Urban Construction, Pingdingshan 467036, China; 5Department of Ecology and Environment of Henan Province, Zhengzhou 450046, China; weipeng_henan@outlook.com; 6Chongqing Key Laboratory of Materials Surface & Interface Science, Chongqing University of Arts and Sciences, Chongqing 402160, China; xmeng@cqwu.edu.cn (X.M.); cheng20120027@hotmail.com (J.C.)

**Keywords:** bamboo-based biochar, hydroxyapatite, heavy metals, pharmaceuticals, sorption behavior, mechanisms

## Abstract

From the perspective of treating wastes with wastes, bamboo sawdust was integrated with a hydroxyapatite (HAP) precursor to create engineered nano-HAP/micro-biochar composites (HBCs) by optimizing the co-precipitated precursor contents and co-pyrolysis temperature (300, 450, 600 °C). The physicochemical properties of HBCs, including morphologies, porosities, component ratios, crystalline structures, surface elemental chemical states, surface functional groups, and zeta potentials as a function of carbonization temperatures and components of precursors, were studied. Biochar matrix as an efficient carrier with enhanced specific surface area to prevent HAP from aggregation was desired. The sorption behavior of heavy metal (Cu(II), Cd(II), and Pb(II)) and pharmaceuticals (carbamazepine and tetracycline) on HBCs were analyzed given various geochemical conditions, including contact time, pH value, ionic strength, inferencing cations and anions, coexisting humic acid, and ambient temperature. HBCs could capture these pollutants efficiently from both simulated wastewaters and real waters. Combined with spectroscopic techniques, proper multiple dominant sorption mechanisms for each sorbate were elucidated separately. HBCs presented excellent reusability for the removal of these pollutants through six recycles, except for tetracycline. The results of this study provide meaningful insight into the proper integration of biochar–mineral composites for the management of aquatic heavy metals and pharmaceuticals.

## 1. Introduction

The development of industry imposes an extremely negative influence on the aquatic environment. The heavy-metal-containing wastewater discharged from industries, including mining operations, tanneries, and metal plating facilities, directly or indirectly poses a serious threat to the surface and groundwater [1,2,3]. The accumulation of heavy metals in living organisms, even at trace levels, leads to long-term damage to humans, inducing a high risk of diseases and disorders such as cancer, liver and kidney damage, and hypertension [4,5]. The maximum permissible concentrations of Cu, Cd, and Pb in drinking water are 2.0, 0.003, and 0.01 mg/L, according to the World Health Organization [6]. Meantime, the annual consumption of pharmaceuticals per capita has been about 15 g in the world and 50–150 g in industrialized countries during the last decade [7,8,9]. The direct discharge of pharmaceutical wastewater and the partial treatment of wastewater streams from pharmaceutical production’s release of pharmaceutical wastewater into the aquatic environment, and there is great concern about this practice given this wastewater’s potential harm. Among pharmaceuticals, carbamazepine (CBZ) is a typical antiepileptic drug used to treat epileptic seizures and nerve pain, with more than 1000 tons of annual consumption and 30 tons of inevitable release per year [10]. Tetracycline (TC) is classified as an emerging pollutant given its antimicrobial resistance and endocrine-disrupting effect. Around 80% of TC is excreted as a parent compound via urine and feces into the environment [11]. The residues of CBZ and TC induce multi-resistant genes in microorganisms, causing adverse ecological and health impacts. Therefore, it is highly desired to remove heavy metals and pharmaceuticals from aqueous solutions. Among various approaches, the sorption technique can dispose of pharmaceutical-containing wastewater and has continually attracted attention because of its wide adaptability, easy operability, and economical favorability [12,13,14,15,16,17,18]. So far, various natural and artificial materials have been extensively exploited to purify contaminated wastewater. However, many sorbents are restricted due to their low sorption capacity, weak stability, and narrow availability [19]. During the last twenty years, significant attention has been focused on the preparation of advanced materials, including metal–organic frameworks, covalent organic frameworks, zero-valent irons, and carbonaceous nanomaterials [20,21,22,23]. For instance, activated carbon for the remediation of Pb(II), carbon fibers for the enrichment of As(V), carbon nanotubes for the uptake of Cr(VI), and graphene oxides for the capture of TC have been reported with high sorption capacities [24,25,26,27], whereas when taking a serious note of both the high cost of advanced artificial materials and the content complexity of actual wastewater, multifunctional sorbents to capture multiple pollutants are highly important.

According to the State Forestry Administration of China, the production of bamboo in China was about 18 million tons each year planted in an area of around 6 million hectares [28]. Accordingly, the random treatment of huge amounts of bamboo wastes, including via simple landfills or open burning, would induce serious pollution. As a carbon-rich solid produced from the pyrolysis of waste residuals in oxygen-limited conditions, biochar is able to capture positively charged metal species and organic waste molecules due to its intrinsic characteristics, such as negatively charged surfaces in aqueous solutions and aromatic components in a matrix. The application of biochars with high efficiency in the remediation of pollutants in waters or soils would provide a promising alternative for the resourceful utilization of biowaste residuals. Given the significantly lower costs of biochars (around USD 51–386/ton) compared to commercial activated carbon (around USD 3000/ton), the usage of biochars would be highly desired from the view of waste utilization and environmental sustainability [29]. However, pristine biochars prepared without modification are still impeded by their intrinsic shortcomings, such as the relatively large particle size, the smaller exposed surface area, and the scarcity of surface functionality. To extend the wide usage of pristine biochars with improved capture capabilities, physical (e.g., using the steam of oxidized gas, plasma, and microwave), chemical (e.g., pre-/post-treatment using acids, alkalis, metal salts), and composite (e.g., via loading micro-/nano-particles) activation have been carried out [30,31,32]. The combination of biochar with some known sorbents is more attractive given its advantage of facile preparation, versatility, and significant enhancement of uptake capacity. Eco-friendly hydroxyapatite (HAP) with low manufacturing costs has presented the capability of removing heavy metals through precipitation, ion exchange, and surface complexation [33]. Biochar and HAP both exhibited a favorable sorption capability for aquatic pollutants. Nevertheless, the easy self-aggregation of HAP to the size of a micrometer and even millimeter to reduce the high surface energy greatly limits its broad application. Given the characteristics of biochar and HAP, the ability to create biochars with an appropriate surface area and functional uptake sites as a carrier to prevent HAP from aggregation is highly anticipated. Since the biochar could be integrated with HAP through a co-precipitation approach [34], the optimization of biomass/HAP precursor ratios and pyrolysis temperature to achieve the efficient integration of nano-HAP and a micro-biochar matrix is crucial for the preparation of this kind of composite sorbent.

In this work, we aimed to (1) optimize the integration of nano-HAP/micro-biochars (HBCs) via the pyrolysis of bamboo sawdust co-precipitated with a HAP precursor of various mass ratios under three heating temperatures; (2) investigate the physicochemical properties of HBCs, including the morphology, porosity, component ratio, crystalline structure, surface elemental composition and state, functional group, and surface zeta potentials with carbonization temperatures and component ratios of precursors; (3) study the sorption behavior of heavy metal (Cu(II), Cd(II), and Pb(II)) and pharmaceuticals (carbamazepine (CBZ) and tetracycline (TC)) on HBCs under various geochemical conditions (i.e., contact time, pH value, ionic strength, interfering cations and anions, coexisting humic acid (HA), and ambient temperature) in ideal, simulated, and real waters; (4) elucidate the underlying mechanisms for the sorption of heavy metals and pharmaceuticals on HBCs combining the spectroscopic analysis.

## 2. Material and Methods

### 2.1. Materials and Reagents

Since Guanghan (Sichuan China) is an important source of bamboo products, bamboo sawdust was collected directly from a local bamboo product plant. Salts of heavy metals and other chemicals were purchased from Aladdin Chemistry Co., Ltd. (Shanghai China). All chemicals and reagents were of analytic grade without further purification. Deionized (DI) water with a resistivity of 18.2 MΩ cm (Milli Q plus, Merck Millipore Co., Darmstadt, Germany) was applied in the experiments.

### 2.2. Preparation of Hydroxyapatite-Modified, Bamboo-Based Biochar

Bamboo sawdust was crushed to pass through a 200-mesh sieve. The schematic for preparing HBCs is illustrated in Figure 1a. A certain amount of bamboo sawdust (1.0, 2.0, and 3.0 g) was added with 5.0 mL CaCl_2_ (2.0 mol/L) with violent agitation and ultrasound of 3 h. Then, 5.0 mL (NH_4_)_2_HPO_4_ (1.2 mol/L) was mixed with the above suspension vigorously with pH values of 10.0–10.5. After being aged for one day, the solid precipitate was washed thoroughly to neutral and was pyrolyzed in a tube furnace at temperatures of 300 °C, 450 °C, and 600 °C with a continuous N_2_ flow of 200 mL/min for 2 h. After being cooled to room temperature, HBCs with various contents of HAP and biochar were ground and passed through a 200-mesh sieve. The obtained HBCs were termed as HBCx-y, where x is the weight of bamboo sawdust added, and y is the pyrolysis temperature. For comparison, pure HAP was prepared by the same procedures without the addition of bamboo sawdust.

### 2.3. Characterization of HBCs

The morphology was recorded using scanning electron microscopy (SEM) (Carl Zeiss NTS GmbH, Oberkochen, Germany). The crystalline structures of HBCs before and after the sorption were determined via X-ray diffraction (XRD, Bruker Co., Billerica, MA, USA) with CuK-α radiation at 40 kV (λ = 0.15418 nm). Thermal gravimetric analysis (TGA) was taken by a Shimadzu TGA Q5000 V3.15 Build 263 thermogravimetric analyzer with an air flow rate of 75.0 mL/min. The surface functional groups for HBCs before and after sorption were measured by Fourier transform infrared spectroscopy (FTIR, Perkin-Elmer, Waltham, MA, USA). The specific surface area and porosity were measured using an Autosorb-1C (Quantachrome Instruments, Boynton Beach, FL, USA) at 77 K with a degassing for 8.0 h at 200 °C. X-ray photoelectron spectroscopy (XPS) was recorded on a VG Scientific ESCALAB Mark II spectrometer equipped with two ultrahigh vacuum (UHV) chambers. Zeta potentials of HBC surfaces in the pH range of 2.0–8.0 were analyzed via a Zetasizer Nano ZS90 Analyzer (Malvern, UK).

### 2.4. Batch Experiments and Data Analysis

Batch experiments were performed in conical flasks oscillated at a frequency of 200 rpm for 24 h. The effects of geochemical conditions include contact time (0.0–1440.0 min), pH (2.0–7.0), ionic strength (0.01–0.20 mol/L), HA molecule (0–30 mg/L), and temperature (293–313 K) were studied, respectively. All experimental data were the averages of triplicate determinations, and the relative errors of the data were less than 5.0%. The sorption of pollutants in simulated wastewaters and real waters was also performed. HBCs were regenerated using 0.1 mol/L of KH_2_PO_4_ solution and trichloromethane, respectively. Detailed information on the sorption experiments and analytical methods is shown in the Appendix A.

## 3. Results and Discussion

### 3.1. Characterization of HBCs

The morphologies of HBCs were long strips with irregular geometries (Figure 1b–m). As the magnification rose to 20,000×, plenty of nano-sized particles were agglomerated on the surface of the micro-sized biochar matrix, which were presumably the loading nano-HAPs. As the mass content of bamboo sawdust increased (Figure 1c,e,g), the agglomeration degree of HAP decreased gradually with an increased surface of exposed biochar. HAP particles were partially embedded into or blocked the pores of biochar with the rising pyrolysis temperature (Figure 1g,i,k), which was analogical to the reported engineered magnetic biochar derived from marine macroalgal biomass [35]. The micro-biochar matrix exerted as a carrier to protect HAP from self-aggregation efficiently, achieving the integration of micro-nano-engineered HBCs. Meanwhile, the sizes of HAP prepared without biochar carrier were far above the nanoscale.

Both the total surface area (SA_total_) and microporous surface area (SA_micropore_) of HBCs were obviously increased with the mass ratio of bamboo sawdust in the precursor, i.e., 62.53, 86.95, and 102.75 m^2^/g for SA_total_ of HBC1-600, HBC2-600, HBC3-600, and 23.94, 34.94; and 51.82 m^2^/g for SA_micropore_ of HBC1-600, HBC2-600, HBC3-600 (Figure 1n). SA_total_ and SA_micropore_ significantly increased from 7.68 to 102.75 and from 0 to 51.82, respectively, for HBC3-300 to HBC3-600 (Figure 1o). The release of volatile matter from bamboo sawdust during the carbonization would generate newly formed micropores with their evolvement to macro-/meso-pores. The calculated pore volume through the BJH method also presented a rising tendency with the pyrolysis temperature from 300 °C to 600 °C. Thermogravimetric profiles of HBCs (Figure 1p,q) showed a two-step weight loss of composites. The initial weight loss for five samples from 40 °C to 150 °C reflected the loss of moisture. The second weight loss from 150 °C to 530 °C was attributed to the loss of CO and/or CO_2_ from the decomposition of O-containing groups and the complete combustion of carbon backbones of biochars [36]. Specifically, the starting temperatures to lose weight were around 150 °C, 200 °C, and 290 °C for HBC3-300, HBC3-450, and HBC3-600, respectively. Since HAP has excellent thermal stability, the mass ratios of HAP in dry HBCs (not including moisture) were calculated to be 77.33% (HBC1-600), 61.96% (HBC2-600), 45.19% (HBC3-600), 34.24% (HBC3-450), and 17.64% (HBC3-300).

Six typical diffraction peaks located at the 2θ of 25.8°, 31.8°, 39.6°, 46.7°, 49.6°, and 53.2° (Figure 2a,c) were ascribed to the (002), (211), (310), (222), (213), and (004) reflections of HAP (JCPDS PDF 00-001-1008), indicating the successful formation of HAP particles on HBCs [37]. The enlarged patterns of HBCs displayed broad peaks located from 7.5° to 35.0°, which were ascribed to the amorphous carbon in HBCs. The increasing marked length from l3 to l1 (l1 > l2 > l3 marked in Figure 2b indicated that the content of amorphous carbon increased from HBC1-600 to HBC3-600, which is consistent with the rising ratio of bamboo sawdust in the precursor. The diffraction peak at 25.80° (HBC3-600) shifted to 26.04° (HBC3-450) and 26.15° (HBC3-300), which was due to the increasing degree of crystallinity of HBCs. Moreover, the crystalline structure of HAP was not damaged by heating in the N_2_ atmosphere, verifying the excellent stability of HAP during thermal disposal.

Typical C, O, N, Ca, and P elements were presented in the survey XPS spectra of HBCs (Figure 2d). Specifically, C 1s spectra (Figure 2e) were deconvoluted into four components, including C-C at 284.8 eV, C-O/sp^2^ C-N at 286.0 eV, C-O/sp^2^ C-N at 287.6 eV, and O-C=O at 289.6 eV [38]. As the pyrolysis temperature increased from 300 °C to 600 °C, the C-C ratio decreased from 88.29% to 77.32%, while the rising ratios of both C-O/sp^2^ C-N and C-O/sp^2^ C-N should not be ignored. Given the difficulty of deconvoluting O elemental species into carboxyl, carbonyl, ethers, and inorganic O in -PO_4_, O 1s spectra (Figure 2f) in this study were divided into two components at 530.9 eV (C-O and P-O), and 532.9 eV (C=O and P=O), respectively [38]. The ratio of single-bonded O declined from 94.84% (HBC1-600) to 83.02% (HBC3-600) due to the existence of double-bonded O in the biochar matrix. The double-bonded O was not observed on the surface of HBC3-450 and HBC3-300, given their limited pyrolysis temperatures. Moreover, N 1s spectra (Figure 2g) were deconvoluted into pyridinic-N at 398.2 eV, amino-N at 399.2 eV, pyrrolic-N at 400.4 eV, and quaternary-N at 401.7 eV [39]. These chemical states of N probably played various roles in the sorption of aquatic pollutants such as heavy metals and pharmaceuticals.

FTIR spectra of HBCs (Figure 2h) presented broad bands between 3100 cm^−1^ and 3700 cm^−1^ corresponding to the stretching vibrations or attached moisture of the -OH group. Bands at 2926 cm^−1^ and 2851 cm^−1^ were ascribed to the C-H stretching vibrations of the methylene group. The intensities of -CH_2_-CH_3_ peaks declined gradually with the pyrolysis temperature from 300 °C to 600 °C, verifying the improving aromaticity of biochar species in HBC(1-3)-600 [40]. The stretching vibrations of aromatic -C=O and -C-O at 1612 cm^−1^, 1451 cm^−1^, and 1034 cm^−1^, respectively, were presented. The ν_2_ O-P-O and ν_4_ O-P-O bending vibrations located at 602 cm^−1^ and 566 cm^−1^, respectively, were also clearly observed, corresponding to the tetrahedral -PO_4_^3−^ groups of HAP component in HBCs [40]. The enhancement of pH_zpc_ (zero point charge) (i.e., 2.59 mV for HBC3-300, 2.78 mV for HBC3-450, and 3.07 mV for HBC3-600, Figure 2i) indicated the decline in surface O-containing groups with increasing pyrolysis temperatures. In general, the negatively charged surfaces of HBCs would be more available to capture positively charged heavy metals in the specific pH range. Moreover, the zeta potential would also increase (e.g., 3.07 mV for HBC3-600, 3.28 mV for HBC2-600, and 3.39 mV for HBC1-600) as the mass content of the HAP component rose from 41.19% to 77.33% in HBCs.

### 3.2. Sorption of Cu(II), Cd(II), and Pb(II)

#### 3.2.1. Sorption Kinetics

The uptakes of Cu(II) and Cd(II) on HBCs were composed of two stages (Figure 3a,b), i.e., *q_t_* (the real-time sorption amount of metal ions on each gram of sorbent) increased distinctly in the first period of 240 min, followed with an equilibrated stage with almost constant *q_t_*. Different from those of Cu(II) and Cd(II), *q_t_* of Pb(II) on HBCs reached their maximum values at 90 min before achieving equilibrium (Figure 3c). The relatively higher diffusion resistance of heavy metals to be adsorbed to the interior channels of HBCs would induce the sorption process more slowly before arriving at the equilibrium point. As for the simulated parameters fitted using the pseudo-first-order model (Appendix A) and pseudo-second-order model (Appendix A), the coefficients R^2^ listed in Appendix A indicated that the pseudo-second-order model was more appropriate to describe the sorption kinetics. Hence, it was suggested that the sorption should be primarily controlled by rate-limiting chemisorption rather than physical sorption [41].

#### 3.2.2. Effects of Solution pH, Ionic Strength, and Coexisting Cation, Anion, and HA

*q_e_* of Cu(II), Cd(II), and Pb(II) rose obviously with the increasing pH from 2.0 to 7.0 (Figure 3d–f). Since the dominant species of these heavy metals in aqueous solutions were positively charged or hydrated ions in pH 2.0–7.0 (e.g., Pb^2+^, PbOH^+^, Pb_3_(OH)_4_^2+^, Pb_4_(OH)_4_^4+^, Cu^2+^, Cu_2_(OH)_2_^2+^, Cu_3_(OH)_4_^2+^, and Cd^2+^), the positively charged surfaces of HBCs induced by the protonation would electrostatically repulse these positively charged heavy metals at pH < pH_pzc_. Meanwhile, the negatively charged surfaces of HBCs induced by the deprotonation would provide more uptake sites to electrostatically attract positively charged heavy metals at pH > pH_pzc_, enabling the increasing *q_e_*. The ionic strength of NaNO_3_ ranging from 0.01 to 0.50 mol/L presented a significant influence on *q_e_* of Cu(II), Cd(II), and Pb(II) at pH 5.0 (Figure 3g–i). The concentration of Na^+^ ions can change the binding species by influencing the double electrode layer thickness and interface potential. The steric hindrance of Na^+^ ions would increase with ionic strength, resulting in the decline in *q_e_* of Cu(II), Cd(II), and Pb(II). Since the outer sphere surface complexation and cation exchange of sorbates with sorbents is much more sensitive to ionic strength than inner sphere surface complexation, the electrical double layer complexes formed between heavy metals and HBCs were conducive to the sorption [42]. It is suggested that ion exchange interaction or outer sphere surface complexation was the dominant mechanism in the sorption of Cu(II), Cd(II), and Pb(II) on HBCs.

The interference of common cations and anions on the adsorption of Cu(II), Cd(II), and Pb(II) is shown in Appendix A. Monovalent background cations in the electrolyte solution (e.g., Li^+^, Na^+^, K^+^, Rb^+^, Cs^+^) exerted a slight influence on the adsorption. Meanwhile, the added divalent cations (e.g., Mg^2+^, Ca^2+^, Sr^2+^, Ba^2+^) restricted the adsorption of Cu(II), Cd(II), and Pb(II). In general, the cations with higher valence were much more easily adsorbed by HBCs, and the divalent cations would occupy twice the number of sites more competitively by forming (=SO)_2_–M^2+^. Meanwhile, the ionic radius of the divalent cation is smaller compared to that of the monovalent cation in the same period as follows: Li^+^ (0.76 Å), Na^+^ (1.02 Å), K^+^ (1.38 Å), Rb^+^ (1.52 Å), Cs^+^ (1.67 Å), Mg^2+^ (0.72 Å), Ca^2+^ (1.00 Å), Sr^2+^ (1.18 Å), and Ba^2+^ (1.35 Å) [43,44]. Therefore, given the higher valence and lower ionic radius of divalent cations, Mg^2+^, Ca^2+^, Sr^2+^, and Ba^2+^ were more competitive in binding on the surface sites, decreasing the sorption of Cu(II), Cd(II), and Pb(II). The interference of electrolyte anions on Cu(II), Cd(II), and Pb(II) by HBCs is presented in Appendix A. NO_3_^−^, Cl^−^, and ClO_4_^−^ exerted a negligible influence on the sorption, even though these negatively charged anions may form complexes with O-containing groups on the surfaces of HBCs.

All the surfaces of HBCs were negatively charged at pH 5.0 and would electrostatically repulse the negatively charged HA molecules. Thus, the sorption of Cu(II), Cd(II), and Pb(II) would be limited due to the formation of binary HA-heavy metal complexes. For *q_e_* of Cu(II), Cd(II), and Pb(II) on HBCs as a function of HA concentrations at pH 5.0 illustrated in Figure 3j–l, all *q_e_* values decreased obviously with the rising HA concentrations from 0 to 30 mg/L. For instance, the ratios decreased by 28.6% with the capture of Cu(II) by HBC3-300, 31.6% with the capture of Cd(II) by HBC3-450, and 28.4% with the capture of Pb(II) by HBC3-600.

#### 3.2.3. Sorption Isotherms and Thermodynamic Study

As presented in Appendix A, the sorption isotherms of Cu(II), Cd(II), and Pb(II) on HBCs presented typical L shapes with positive correlations with the temperature. For the obtained fitting coefficients (R^2^) listed in Appendix A, the Langmuir model was more appropriate to fit those isotherms, demonstrating the homogenous distribution of available sorption positions on the surfaces of HBCs and the occurrence of the sorption via a monolayer coverage [45]. The maximum sorption capacities (*q_max_*, Figure 4) of Cu(II), Cd(II), and Pb(II) on HBC3-600 (201.7, 176.5, and 405.7 mg/g) at pH 5.00, T 293 K, and 0.01 M NaNO_3_ were obtained from the Langmuir simulation. The declined order of *q_max_* for Pb(II) > Cu(II) > Cd(II) probably could be attributed to the decreasing order of hydrated radii for Cd^2+^ (4.26 Å) > Cu^2+^ (4.19 Å) > Pb^2+^ (4.01 Å) [46]. Heavy metal ions with smaller hydrated radii commonly presented a higher affinity and attraction force in the sorption process. To evaluate the sorption properties of HBCs toward Cu(II), Cd(II), and Pb(II), their *q_max_* were compared with other reported materials, as shown in Appendix A. *q_max_* values in this study were superior to many sorbents, including minerals, biochars, zero-valent iron nanoparticles, carbon nanotubes, graphene, and layered double hydroxide. HBCs exhibited excellent potential in the elimination of aquatic heavy metals.

Moreover, a higher temperature was conducive to the capture of heavy metals by HBCs. The activity of heavy metals showed a positive relationship with the sorption temperature, and HBCs were induced to produce more available uptake positions to accelerate the intraparticle diffusion of heavy metals into the pores of HBCs. The negative values of ΔG° for the sorption of Cu(II), Cd(II), and Pb(II) on HBCs (Appendix A) were more negative with the temperature between 293 K and 313 K. This demonstrated that the sorption of Cu(II), Cd(II), and Pb(II) on HBCs was a spontaneous process with thermodynamic favorability. The positive ΔH° values indicated the above-mentioned sorption was endothermic. Moreover, the positive ΔS° exhibited increasing randomness of Cu(II), Cd(II), and Pb(II) captured on the solid–liquid interface of HBCs.

### 3.3. Sorption of CBZ and TC

#### 3.3.1. Sorption Kinetics

The uptakes of CBZ on HBCs were composed of two stages (Figure 5a,b), i.e., an obvious increase in *q_t_* in the first contact time of around 120 min before reaching equilibrium. On the contrary, three stages of kinetics were exhibited for the sorption of TC on HBCs. Specifically, *q_t_* of TC rose quickly in the first stage until 180 min, and then the tendency slowed down gradually and achieved equilibrium at around 1200 min. The longer sorption equilibrium time for TC on HBCs should be attributed to the larger molecular size of TC, which had a relatively higher diffusion resistance to surpass to enter the interior porous channels [46]. From the fitted parameters of two pseudo kinetic models (Appendix A), the pseudo-second-order model was more fitting to describe the sorption processes of CBZ and TC on HBCs. Accordingly, the above-mentioned sorption processes should primarily be governed by rate-limiting chemisorption rather than physical sorption [41].

#### 3.3.2. Effects of Solution pH and Ionic Strength

As shown in Figure 5c, *q_e_* values of CBZ were consistent throughout a pH range of 2.0 to 8.0 (around 9.52, 62.9, 79.1, 90.3, and 101.1 mg/g on HBC3-300, HBC3-450, HBC3-600, HBC2-600, and HBC1-600, respectively). CBZ was presented with the neutral charge within this range (pKa of 2.3 and 13.9 for -CONH_2_ group [47,48]. This consistent trend for the sorption of CBZ at a pH of 2.0 to 8.0 could be explained well as being due to a lack of electrostatic interaction between CBZ and positively/negatively charged HBCs. The sorption of CBZ onto HBCs should be controlled by hydrophobic interaction, π–π interaction, and electrostatic interaction. *q_e_* values of TC decreased slightly with pH from 2.0 to 8.0, as displayed in Figure 5d. The dominant anionic form of TC (pKa values of 3.3, 7.7, 9.7, and 12.0) would be electrostatically repulsed by HBCs since the negative charge on the surfaces of HBCs increased gradually from pH 2.0 to 8.0 [49]. π–π interaction and hydrogen bonding should not be ignored for the mechanism of TC sorption, which will be discussed in the following content. *q_e_* of CBZ and TC on HBCs was stable in the presence of various concentrations of NaNO_3_ ranging from 0.01 to 0.50 mol/L (Figure 5e,f), showing the weak influence of ionic strength on the sorption of CBZ and TC. Given the non-sensitivity for ionic strength, ion exchange interaction or outer sphere surface complexation should be the dominant mechanism for the sorption of CBZ and TC on HBCs.

#### 3.3.3. Sorption Isotherms and Thermodynamic Investigation

Similar L shapes were observed in the sorption isotherms of CBZ and TC (Appendix A). HBCs with homogenously distributed sorption sites captured CBZ and TC via a monolayer coverage because the Langmuir model could simulate these isotherms better, as listed in Appendix A. Simulated *q_max_* values of CBZ and TC on HBC3-600 (168.9 and 105.1 mg/g) at pH 6.00, T 293 K, and 0.01 M NaNO_3_ were obtained (Figure 5g). A higher pyrolysis temperature would facilitate the cleavage of weak bonds with the deoxidizing of functional groups through the volatilization of organic fractions of bamboo sawdust, leading to a higher proportion of hydrophobic groups, which was beneficial for the capture of pharmaceuticals. Meanwhile, HBCs pyrolyzed at lower temperatures exhibited lower *q_max_* for CBZ and TC due to their lower surface areas and smaller pore sizes. In addition, given the smaller molecular diameter of CBZ (0.9 nm) compared to TC (1.4 nm), CBZ would diffuse into the pores of HBCs more easily, presenting higher *q_max_* values than TC. Moreover, as the *q_max_* of CBZ and TC on various reported sorbents compared in Appendix A, HBCs were superior to many published materials. The thermodynamic parameters (Appendix A) indicated the spontaneity and thermodynamic favorability of the sorption processes. The positive ΔH° and ΔS° indicated the endothermic sorption with rising randomness on the solid–liquid interface of HBCs.

### 3.4. Reusability and Sorption in Simulated Wastewater

Considering the efficiency of desorption, acid pickling (0.5 mol/L of HCl solution) was performed to regenerate the heavy-metals-sorbed HBCs by providing an excessive amount of H^+^ to replace loaded heavy metals, and trichloromethane washing was carried out to desorb the captured pharmaceuticals on HBCs. After being immersed in the eluting solution and stirred for 24 h, the obtained HBCs were applied to capture these targeting pollutants. *q_e_* of Cu(II), Cd(II), and Pb(II) decreased by less than 30.0% after six rounds of recycling (Figure 6a), indicating the favorable reusability of HBCs. The unrecoverable ratio of *q_e_* was probably due to the formation of a stable phase (e.g., precipitates) of heavy metals in HBCs. Moreover, *q_e_* values of CBZ on HBCs decreased less than 15.0% after six recycling rounds, while those of TC on HBCs declined distinctly with ratios from 72.2% to 85.2% after six cycles. The main part of sorbed TC on HBCs could be desorbed by the washing of trichloromethane, given the stronger attachment of TC with HBCs. The sorption of Cu(II), Cd(II), and Pb(II) on HBCs was conducted in the artificial wastewater with initial pH values of 2.0 and 4.0 (Figure 6b). Taking HBC3-600 as the example, Cu(II), Cd(II), and Pb(II) were removed simultaneously with *q_e_* values of 21.7, 8.0, 37.9 mg/g at pH 2.0 and 44.0, and 10.6, 62.1 mg/g at pH 4.0, respectively. This result proved that HBCs were an excellent sorbent given their favorable uptake efficiencies for multiple coexisting heavy metals in acid conditions. The real waters were taken from the Yazi River (Guanghan Sichuan China), Yellow River in the region of Huiji District (Zhengzhou Henan China), and Yangtze River in the region of Yongchuan District (Chongqing China). The salts of Cu(II), Cd(II), and Pb(II) were added into the real waters separately. The sorption of Cu(II), Cd(II), and Pb(II) in these real waters decreased, given the high salinity in these real waters (Figure 6c).

### 3.5. Potential Sorption Mechanisms

XRD patterns of HBC3-600 presented no obvious difference in the diffraction peak of HBC3-600 before and after the uptake of Cu(II) and Cd(II) (Figure 7a), whereas two peaks appeared at 2θ of 21.47°, 30.92°, and 43.76° in the pattern of Pb(II)-loaded HBC3-600, which corresponded to the typical reflections of (Pb_x_Ca_1−x_)_5_(PO_4_)_3_(OH) (JCPDS PDF 01-087-2477) [50]. The similar ionic radii of Ca(II) (0.99 Å) with Pb(II) (1.20 Å) enabled the replacement of Ca(II) with Pb(II) with the formation of an isomorphic solid (Pb_x_Ca_1−x_)_5_(PO_4_)_3_(OH) [51]. The dissolution–precipitation process could be described as Ca_5_(PO_4_)_3_OH + H_2_O → 5Ca^2+^(aq) + 3PO_4_^3−^(aq) + 2OH^−^(aq), and 5xPb^2+^(aq) + 5(1 − x)Ca^2+^(aq) + 3PO_4_^3−^(aq) + 2OH^−^(aq) → (Pb_x_Ca_1−x_)_5_(PO_4_)_3_(OH) + H_2_O. Surface chemical states of various elements on the surface of HBC3-600 after the sorption were also conducted using XPS (Figure 7b,c). Taking HBC3-600 as an example, the ratio of -O on the surface O decreased from 83.02% (HBC3-600)to 61.78% (Cu(II)-loaded HBC3-600), 77.87% (Cd(II)-loaded HBC3-600), and 72.04% (Pb (II)-loaded HBC3-600), respectively, demonstrating the role of O-donor functional groups in the sorption of heavy metals. The ratio of reductive N heteroatom (pyridinic-N, amino-N, and pyrrolic-N groups) on the surface N declined from 79.46% (HBC3-600) to 66.22% (Cu(II)-loaded HBC3-600), 74.37% (Cd(II)-loaded HBC3-600), and 70.88% (Pb(II)-loaded HBC3-600), separately, suggesting the participation of reductive surface N in the capture of heavy metals. Given these spectral results, the sorption of Cu(II), Cd(II), and Pb(II) should be dominated by multiple mechanisms (Figure 7d), including (a) the cation exchange of heavy metals with inorganic components of HBCs, (b) surface complexation between O-containing groups and reductive N species on the surfaces of HBCs, (c) electrostatic attraction between positively charged heavy metals and negatively charged surface of HBCs at different pH, (d) π–electron interactions between aromatic/graphitic carbon in HBCs and heavy metals, and (e) significant dissolution–precipitation of Pb(II) with a nano-HAP component.

There are several possible mechanisms for the capture of CBZ and TC on HBCs, including hydrophobic interaction, electrostatic attraction, π–π interaction, and coordination bonding. The hydrophobic interaction of CBZ with HBCs was much stronger due to the much higher hydrophobicity of CBZ (2.45 of logK_ow_) than TC (−1.37 of logK_ow_) [52]. The electrostatic attraction of TC with the charged surface of HBCs would be much stronger than that of CBZ, given the distributed species of CBZ (neutral state) and TC (weak anions) in the aqueous solutions. The electron-donating group –NH_2_ and benzene rings in CBZ and TC made them captured by HBCs through the π–π donor-acceptor interaction and π–π stacking interaction, respectively [53]. It is also reasonable for TC to be captured by HBCs through chemical or hydrogen bonding. Therefore, the sorption of CBZ was attributed to hydrophobic interaction and π–π interaction, while the uptake of TC was due to hydrogen bonding, π–π interaction, and electrostatic attraction (Figure 7e).

## 4. Conclusions

In summary, micro-nano-engineered HBC3-600 exhibited a biochar matrix carrier with a superior surface area to load the deposited HAP nanoparticles efficiently, presenting excellent *q_max_* of 201.7, 176.5, 405.7 mg/g for Cu(II), Cd(II), and Pb(II) (pH 5.0, T 293 K); and 168.9, 105.1 mg/g for CBZ and TC (pH 6.0, T 293 K) with excellent reusability, except for TC. HBC3-600 could also remove Cu(II), Cd(II), and Pb(II) efficiently from simulated wastewaters and real waters. The spectroscopic analysis revealed that the formation of (Pb_x_Ca_1−x_)_5_(PO_4_)_3_(OH) precipitate was dominant for the high sorption of Pb(II), while the surface complexation of O-containing groups and reductive N species on the surfaces of HBCs played a crucial role in the sorption of Cu(II) and Cd(II). From the perspective of efficiently treating wastes with wastes, this work provides meaningful insight into the integration of micro-biochar and nano-minerals for the remediation of aquatic pollutants and the recycling of agricultural residues.

## Figures and Tables

**Figure 1 nanomaterials-12-01988-f001:**
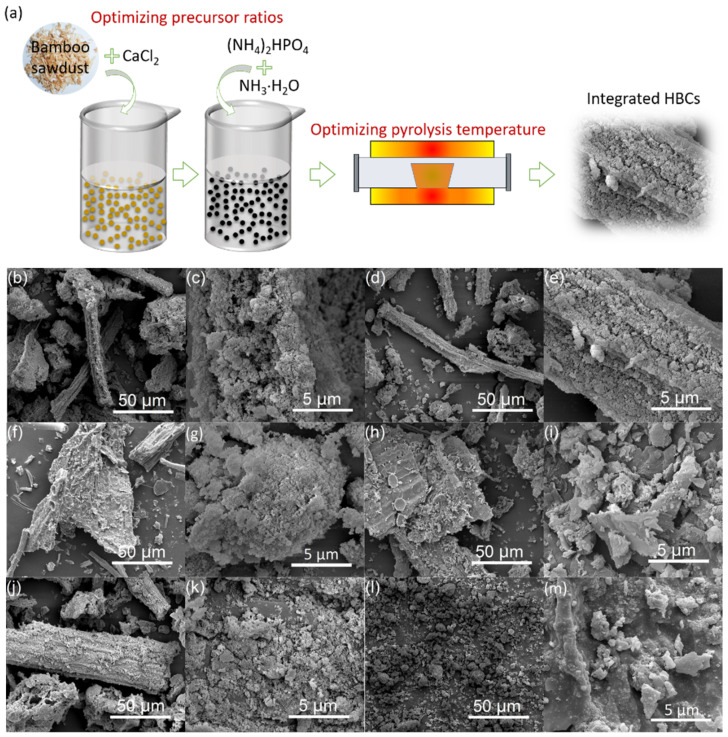
Illustration for preparing HBCs (**a**). SEM images of HBC1-600 (**b**,**c**), HBC2-600 (**d**,**e**), HBC3-600 (**f**,**g**), HBC3-450 (**h**,**i**), HBC3-300 (**j**,**k**), and HAP (**l**,**m**). Low-temperature N_2_ adsorption–desorption isotherms (**n**,**o**) and TGA curves (**p**,**q**) of HBCs.

**Figure 2 nanomaterials-12-01988-f002:**
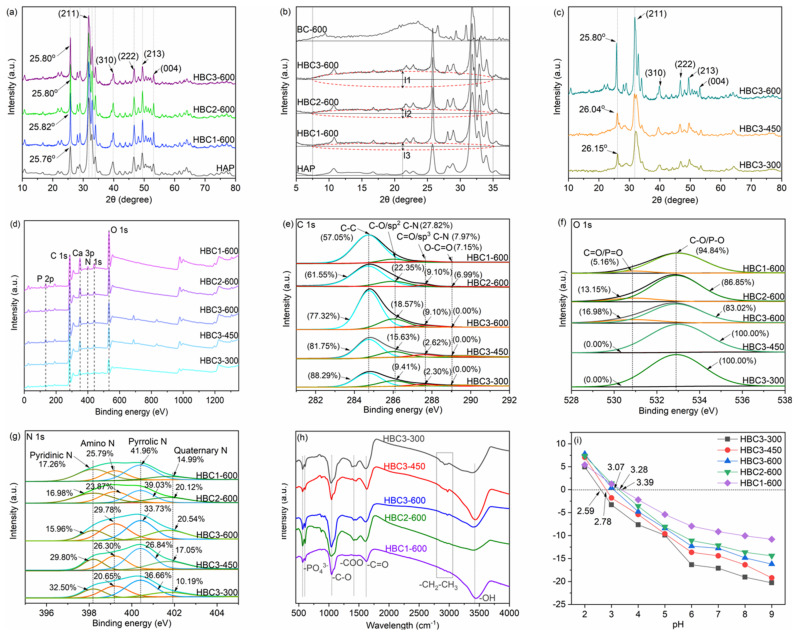
XRD patterns of HBC1−600, HBC2−600, HBC3−600 (**a**,**b**), and HBC3−300, HBC3−450, HBC3−600 (**c**). XPS wide scan spectra (**d**), C 1s spectra (**e**), O 1s spectra (**f**), N 1s spectra (**g**), FTIR spectra (**h**), and zeta potentials (**i**) of HBCs.

**Figure 3 nanomaterials-12-01988-f003:**
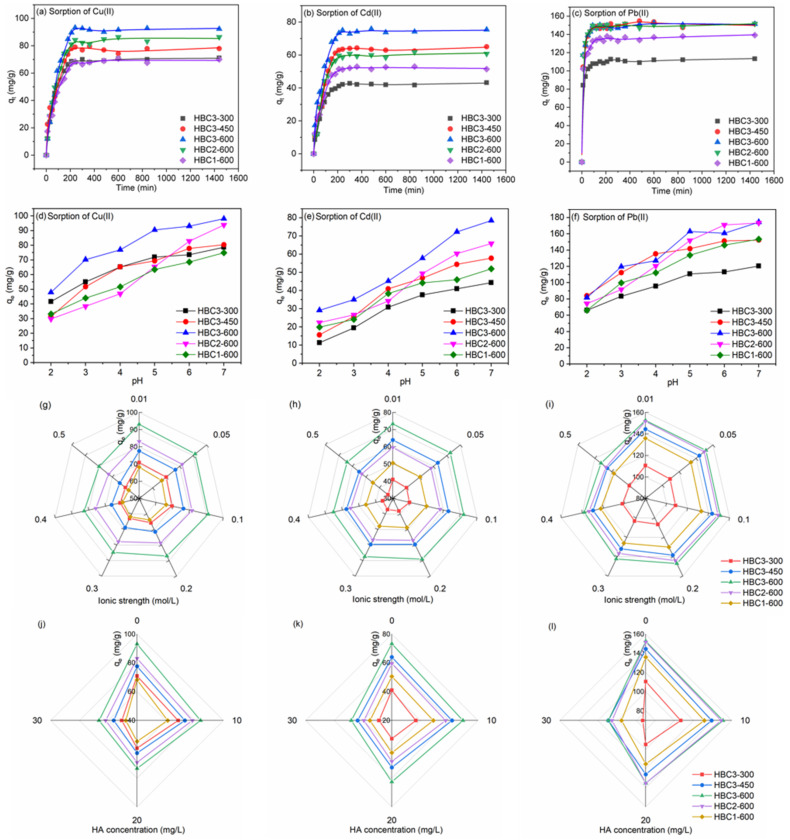
Sorption kinetics of Cu(II) (**a**), Cd(II) (**b**), and Pb(II) (**c**) on HBCs. Effects of pH, ionic strength, and HA on the sorption of Cu(II) (**d**,**g**,**j**), Cd(II) (**e**,**h**,**k**), and Pb(II) (**f**,**i**,**l**) on HBCs. C_[Cu(II)]initial_ = C_[Cd(II)]initial_ = C_[Pb(II)]initial_ = 60.0 mg/L, T = 293 K, m/V = 0.5 g/L.

**Figure 4 nanomaterials-12-01988-f004:**
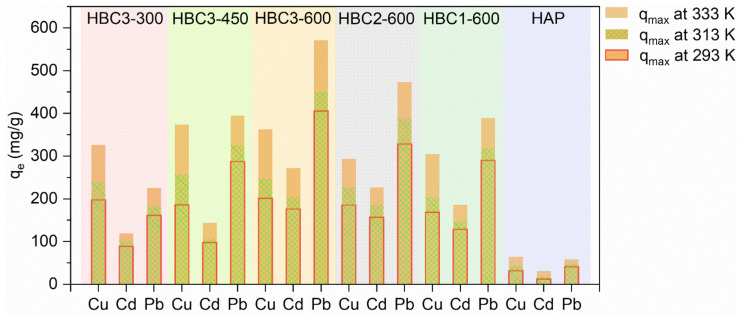
Comparision of *q_max_* of Cu(II), Cd(II), and Pb(II) on HBCs and HAP. pH = 5.0 ± 0.1, T = 293 K, m/V = 0.50 g/L, I = 0.01 mol/L NaNO_3_.

**Figure 5 nanomaterials-12-01988-f005:**
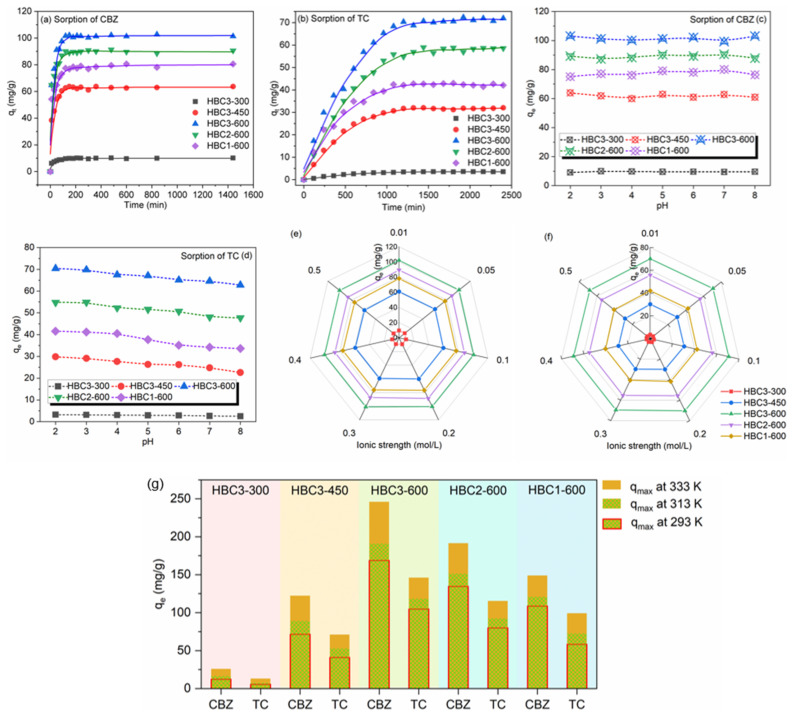
Sorption kinetics of CBZ (**a**), and TC (**b**) on HBCs. Effects of pH and ionic strength on the sorption of CBZ (**c**,**e**), and TC (**d**,**f**) on HBCs. C_[CBZ]initial_ = C_[TC]initial_ = 60.0 mg/L, T = 293 K, m/V = 0.5 g/L. Comparision of *q_max_* of CBZ and TC on HBCs and HAP (**g**) at pH 6.0 ± 0.1, T 293 K, m/V 0.5 g/L, I 0.01 mol/L NaNO_3_.

**Figure 6 nanomaterials-12-01988-f006:**
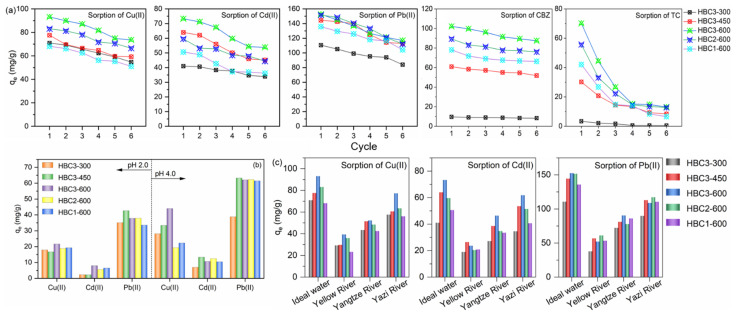
Reusability of HBCs to remove heavy metals and pharmaceuticals from the ideal water (**a**). Sorption of Cu(II), Cd(II), and Pb(II) from artificial wastewater (**b**) and real waters (**c**).

**Figure 7 nanomaterials-12-01988-f007:**
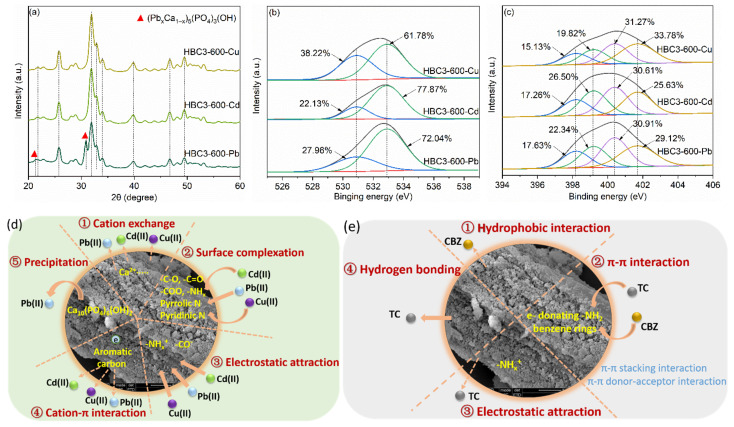
The spectra of heavy-metals-loaded HBC3-600: (**a**) XRD spectra, (**b**) XPS C 1s spectra, (**c**) XPS O 1s spectra. Schematics for sorption mechanisms of Cu(II), Cd(II), Pb(II) (**d**), and CBZ, TC (**e**) on HBCs.

## Data Availability

Not applicable.

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
