# Peer review of "Integration of Micro-Nano-Engineered Hydroxyapatite/Biochars with Optimized Sorption for Heavy Metals and Pharmaceuticals"

_nanomaterials, 2022, doi:10.3390/nano12121988_

Round 1

Reviewer 1 Report

It is a great article but it will requite extensive editing for grammar checks.  Also, several the schematics are difficult to read, especially figure 1, even after 200% magnification. Perhaps authors may rearrange schematics to allow a good magnification and readability compromise.

Reviewer 2 Report

The paper by Zhao et al. is an interesting study on the sorption of various environmental pollutants on synthesized organic-inorganic material. The manuscript contains an unusually large number of results. However, I have a few comments:
1) Conclusions: they seem to contain too much numerical data. What is needed here is a clear indication of which material proved to be optimal for sorption of a given group of pollutants. 
2) Materials and Methods: The sorption process was described in the supplementary materials however a brief description should be given in the main text of the manuscript.
3) The large number of figures is a definite advantage of the article, but it is at the expense of their quality. Especially in the supplementary materials, larger figures could be used.
4) Introduction: It is a bit lost in this part of the article what is new in the particular concept of the material that was decided to synthesize. 

Round 2

Reviewer 1 Report

The authors have expressed some of my concerns.

They have addressed data analysis. They also reconfigured several figures (not all).

Minor spelling and grammar checks and resizing figures for clarity will be sufficient for this publication's acceptance.

Author Response

We greatly appreciate the reviewer for very detailed review of the manuscript. Reviewer’s comments are very helpful for us to significantly improve the quality of the manuscript. Since the reviewer has suggested that our manuscript should undergo the checks of spelling and grammar and the adjustment of figure sizes, we has performed the revision of manuscript accordingly.

Reviewer 2 Report

Thank you for your kind responses and improvements introduced. I feel that the manuscript is ready for publication.

Author Response

We greatly appreciate the reviewer for a very detailed and thorough review of this manuscript. Reviewer’s comments are very helpful for us to significantly improve the quality of the manuscript.